# Female mentors positively contribute to undergraduate STEM research experiences

**Saili Moghe** *, Katelyn Baumgart, Julie J. Shaffer, Kimberly A. Carlson

Department of Biology, University of Nebraska at Kearney, Kearney, Nebraska, United States of America

* moghes1@unk.edu

## Abstract

The positive influence of undergraduate research and mentoring on student success in STEM fields has been well-established. However, the role that the gender of a research mentor may play in the undergraduate research experience warrants further investigation. This is an especially critical issue to address, since the lack of female role models in STEM fields is acknowledged as an impediment to the success and progress of women pursuing STEM-careers. To evaluate how the gender of undergraduate research mentors influences the research experience of students, we collected and analyzed surveys from undergraduates and alumni who had completed undergraduate research at the University of Nebraska at Kearney. We found that even though students did not select mentors based on gender, there were differences in how students perceived their mentors, depending on the gender of their mentors. Interestingly, students with female mentors were more likely than students with male mentors to report that their research experience had prepared them for a career in science. Further, our gender-pairing analyses revealed that students who expressed that the gender of their mentor had contributed to their relationship with their mentor were more likely to have a female mentor. Our data indicate that female mentors favorably influence the undergraduate research experience of both male and female students. Finally, our study reinforces the conclusions of previous studies demonstrating that undergraduate research and mentoring are beneficial for students. Overall, our findings support that, for students to fully benefit from their undergraduate research experience, undergraduate research opportunities for students should include an equitable representation of female mentors.

## Introduction

Undergraduate research benefits students in multiple areas. Past survey studies show that students who have completed undergraduate research achieve better grades [1], are more confident, have a better understanding of science and how research is conducted, as well as a greater awareness about the culture of research [2–4]. Students who engaged in undergraduate research also have a greater interest in Science, Technology, Engineering, and Mathematics (STEM) careers and in obtaining a graduate degree, as compared to those students who did not participate in undergraduate research [2, 5]. Undergraduate research paired with

**Data Availability Statement:** All relevant data are within the manuscript and its Supporting Information files.

**Funding:** The authors received no specific funding for this work.

**Competing interests:** The authors have declared that no competing interests exist.

mentoring is proposed to provide even greater benefits for students in areas of both personal and academic development [6].

In addition to mentoring having positive effects on the career decisions and professional development of undergraduate students [7, 8], mentoring relationships in an undergraduate setting may be an avenue to encourage more women to pursue and succeed in STEM careers [9]. Currently, men are more widely acclaimed in STEM fields than women [10], and outnumber women in science [11], especially in senior positions [12]. This gender disparity in STEM fields [11–13] may mean that aspiring female scientists lack female role models. This creates challenges for women pursuing STEM careers, since they may interpret the paucity of women in STEM as an indication that jobs in STEM are not meant for them [14]. Studies show that starting from an early age, when girls are socially exposed to other girls interested in STEM, they are more likely to maintain their own interests in STEM [15]. Further, characteristics of undergraduate institutes, such as the proportion of women in faculty and leadership positions, also shape how undergraduate women perceive different majors and traditional gender stereotypes [16].

As part of understanding how mentoring relationships can benefit undergraduates, it is important to determine if the gender of a mentor or student might influence the mentor-mentee relationship. Previous studies suggest that both men and women have the potential to harbor gender biases [17, 18]. These studies showed that female students or prospective employees were viewed as less competent and less worthy than their male counterparts and were consequently offered a lower pay or less career mentoring. Therefore, it is necessary to explore the effect that the gender of a student or their mentor has on student success, especially that of female students.

To ensure that students succeed in science careers after graduation, the University of Nebraska at Kearney (UNK) implemented a mentoring program. This program involves students carrying out a research project to help them acquire the experience of working in the scientific field, and to educate them on issues such as the existing gender gap in STEM careers. In addition to making students competitive for future science careers, requiring students to participate in mentor-guided research is expected to prepare them for possible gender-related issues that they might encounter in their future science careers. The purpose of our study was to learn how effective the UNK undergraduate research program has been in enabling students to succeed in the sciences, post-graduation. Additionally, our study aimed to determine if there were any differences in the research experience, including mentoring relationships, of students based on their own gender and their mentor's gender.

## Methods

### Department history and university demographics

Since 1988, the UNK Biology department has included undergraduate research within its degree programs through coursework that involves faculty-mentored student research. This coursework includes 2 courses: 1) BIOL 375 in which students design a research project and select a research mentor and 2) BIOL 420 in which students conduct their designed research under the guidance of their research mentor. Starting from 2010, the program incorporated a hostile work environment training component to address hostile work situations that students may face during the course of their future careers. At the start of the program in 1988, the department had 13 faculty members that included a sole female faculty member, following which female representation increased to 6 female faculty out of a total of 18 faculty members by 2003. The number of female faculty steadily increased to 10 out of 22 faculty members in 2011 and has maintained approximately this proportion since. The UNK undergraduate student population currently consists of 60.8% women and 39.2% men; the population of women

has consistently risen since 2006 at which time 54.2% of the student population was female. Since 2006 until present, the majority (over 75%) of the undergraduate student body is white [19].

## Study details

This study received IRB approval (#090210–1) and began in the Fall of 2010. Surveys were collected until Fall 2016. The Biology Department at UNK asked students who had chosen their research mentor in BIOL 375 (a course in which a student pairs with a mentor and designs a research project), to complete Part I and II of the survey, which asked questions about their demographics and about mentor selection. Students who had completed BIOL 420 (a course in which students conduct their research after being paired with a mentor in BIOL 375), completed Parts I, II and III of the survey, which asked about their undergraduate research experience. To limit social desirability, the survey was completely anonymous with no identifiers. In addition, the survey was worded such that the respondents could not be identified in any way and so that the questions would not be leading. To minimize recall bias, the same survey was given to each participant, and undergraduates completed the survey during the same semester that they were enrolled in the research series. In Fall 2016, UNK alumni who had graduated and had successfully completed the series of undergraduate research classes (BIOL 375 and BIOL 420) were emailed the entire survey, which along with Part I-III, also included Part IV, a final section asking about career outcomes and the influence that undergraduate research has had on their current positions. Alumni included students who completed the program when it first began in 1988. S1 Appendix includes the entire survey (Part I-IV) which was constructed by the Biology department faculty based on questions that the department wanted to gather student responses on. The survey was independently validated by the UNK IRB and by faculty from the Psychology department at UNK who had expertise in analyzing validity, bias, and reliability in surveys. After the surveys were collected, the responses were entered into Excel spreadsheets. Data from respondents of each question were collated (S1 File) and analyzed by the Fishers Exact Test using SPSS Statistics Version 26 software (IBM). The Fisher's Exact Test was used because the sample size of groups being compared was relatively small (less than 1000), and therefore a test that uses an exact procedure was needed, rather than a test such as a Chi-squared test that assumes a large sample size and therefore assumes approximation. A value of $p < 0.01$ was considered as statistically significant.

## Sample description

Survey respondents were either undergraduate students or alumni that were 18 years old or older. There were 484 respondents, of which 304 were undergraduate students and 180 were alumni (S1 File). The respondents reflected similar characteristics as the UNK student body population (as described in the Department history and demographics section). Of the undergraduate respondents, 54.3% were female and 40.6% were male students and of the alumni, 59.4% were female alumni and 40.6% were male alumni. The majority (over 95%) of survey respondents identified as Caucasian in the survey. Surveys were emailed to 1163 alumni, and undergraduate students enrolled within the research series from 2010 to 2016, were provided the opportunity to complete the surveys. The response rate for alumni was 15.5% and 84.7% for undergraduate students (S1 Table).

## Ethics statement

We obtained institutional review board (IRB) approval (#090210–1) from the University of Nebraska at Kearney IRB committee for delivering surveys. The forms were anonymous with

no personal identifiers. We recorded the responses from the surveys into spreadsheets as a number and proceeded with analysis.

## Results

### Mentor selection and mentor assignment

Responses from alumni and undergraduate students combined showed that the majority (94.8%, n = 481) felt that the gender of their mentor did not influence the selection of their undergraduate research mentor. Similarly, most respondents (92.1%, n = 483) stated that gender does not need to be considered when choosing a mentor. When asked about whether male mentors should mentor female students and female mentors should mentor male students, 53.4% (n = 470) respondents answered "Yes," and 46.6% answered "No" (Fig 1).

When alumni responses were compared to undergraduate student responses (BIOL 375 and BIOL 420 students combined), there was a significant difference in how they responded to the question of if gender ought to be considered in mentor selection (p<0.001). While most respondents from both groups responded that gender should not be considered in mentor selection, 95.7% (n = 303) of undergraduates compared to only 86.1% (n = 180) of alumni responded that gender should not be considered in mentor selection (Fig 2A). Undergraduates and alumni also significantly differed in how they responded to the question of if male mentors should mentor female students and if female mentors should mentor male students. Over half (64.4%, n = 177) of the alumni responded "Yes," compared to only 46.8% (n = 293) of the undergraduates who responded "Yes" (Fig 2B). There was no significant difference in responses to questions about selection of mentors when responses from male and female stu-

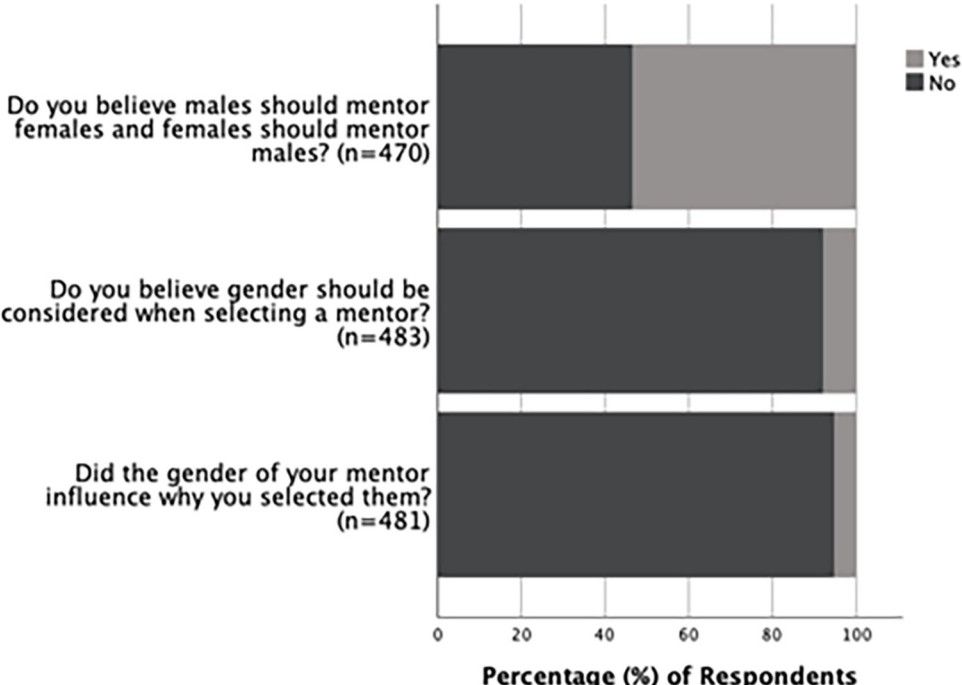

**Fig 1. Views on considering gender when selecting a mentor.** Responses in percentages from alumni and undergraduates regarding their views on considering gender when selecting a mentor.

**(A)**    Do you believe gender should be considered when selecting a mentor?

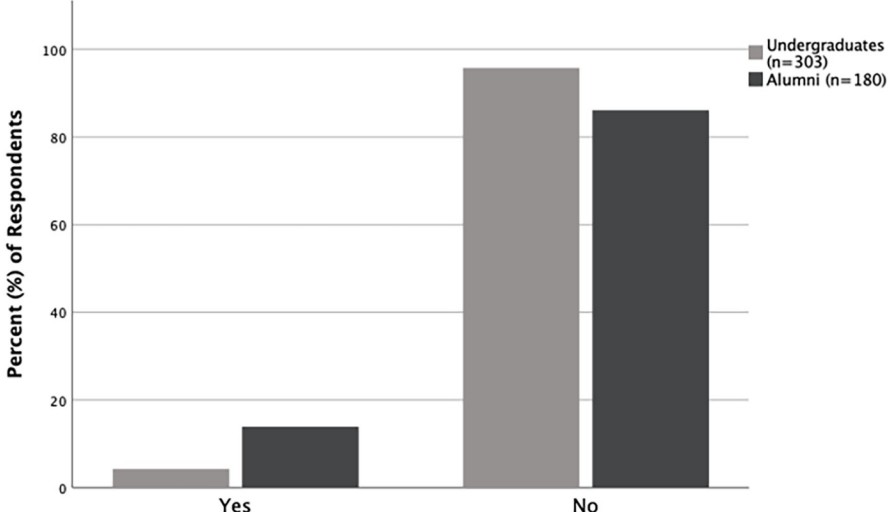

**(B)**    Do you believe males should mentor females and females should mentor males?

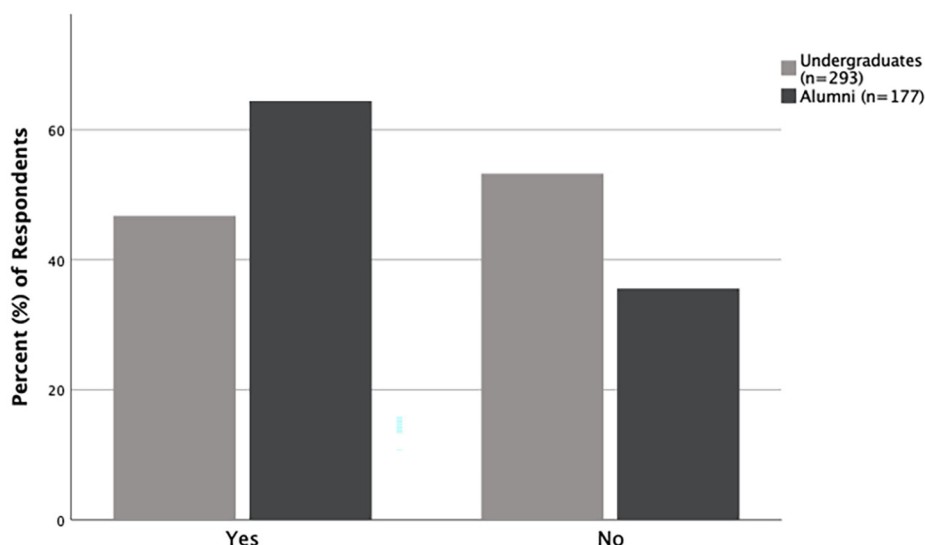

**Fig 2. Differences in alumni and undergraduates regarding their views on gender and mentor selection.** Responses in percentages to questions about gender and mentor selection for which responses from undergraduates and alumni were significantly different (p<0.001). (A) Responses to if respondents believed that gender should be considered during mentor selection. (B) Responses to if respondents believed that male mentors should mentor female students and female mentors should mentor male students.

dents were compared (S2 Table). The assignment of male and female mentors also did not significantly differ by student gender (S3 Table), that is, there was no association between student gender and mentor gender.

## Undergraduate research experience of undergraduate students versus alumni

A large majority of undergraduates and alumni who had completed their undergraduate research responded "Yes" to questions regarding if their overall undergraduate experience was positive and if their mentor was a good role model, helpful, understanding, and available to help (Fig 3). There was no significant difference between undergraduates and alumni in how they responded to most questions about their undergraduate research experience and mentors (S4 Table). However, there was a significant difference (p = 0.001) between how undergraduates and alumni responded to whether their undergraduate experience had prepared them for hostile work environments; 52.2% (n = 134) of alumni, in contrast to only 20.3% (n = 128) of undergraduates, responded "No" to this question (Fig 4).

## Undergraduate research experience of female versus male respondents

A comparison of how male and female respondents (both undergraduate students and alumni) responded to questions about their undergraduate research experience and mentors showed that there was a significant difference in how they responded to if they thought their mentor was a good role model (p = 0.001). Almost 95% of male respondents (n = 135), compared to only 81.6% (n = 179) of female respondents, responded that their mentor was a good role model (Fig 5A). The majority of both genders responded that their undergraduate research experience did not prepare them for work opportunities due to their gender. However, there was still a significant difference between their responses (p = 0.001; Fig 5B). A higher proportion of male respondents (77.8%, n = 108), than that of female respondents (59.6%, n = 141),

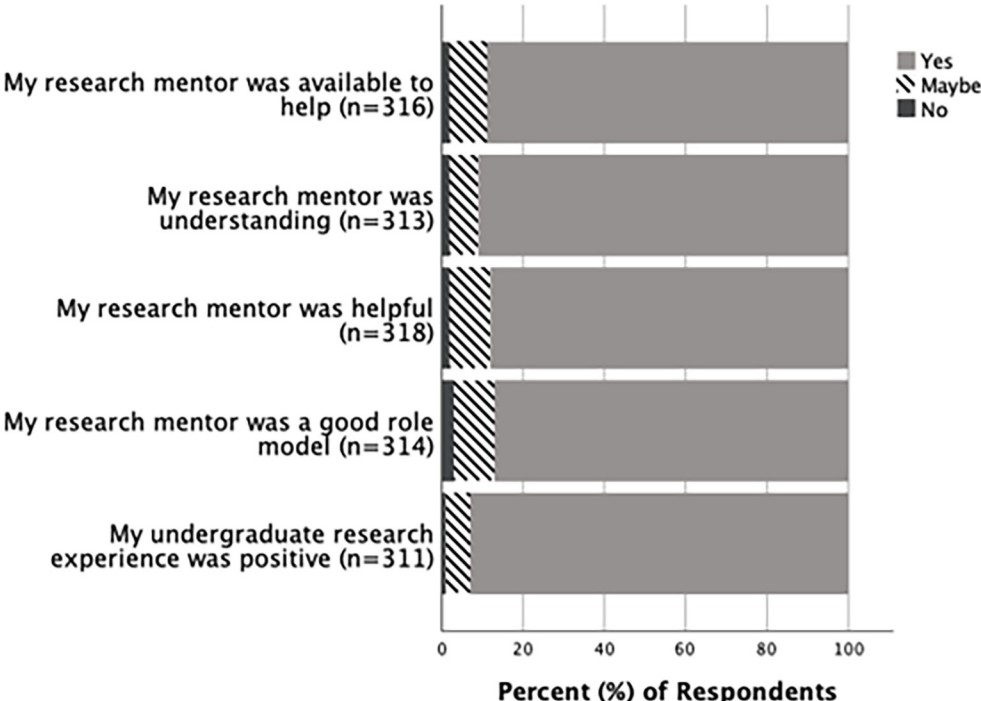

**Fig 3. Undergraduate research experience and perceptions of research mentors.** Responses in percentages from alumni and undergraduates regarding their undergraduate research experience and perceptions of their mentor.

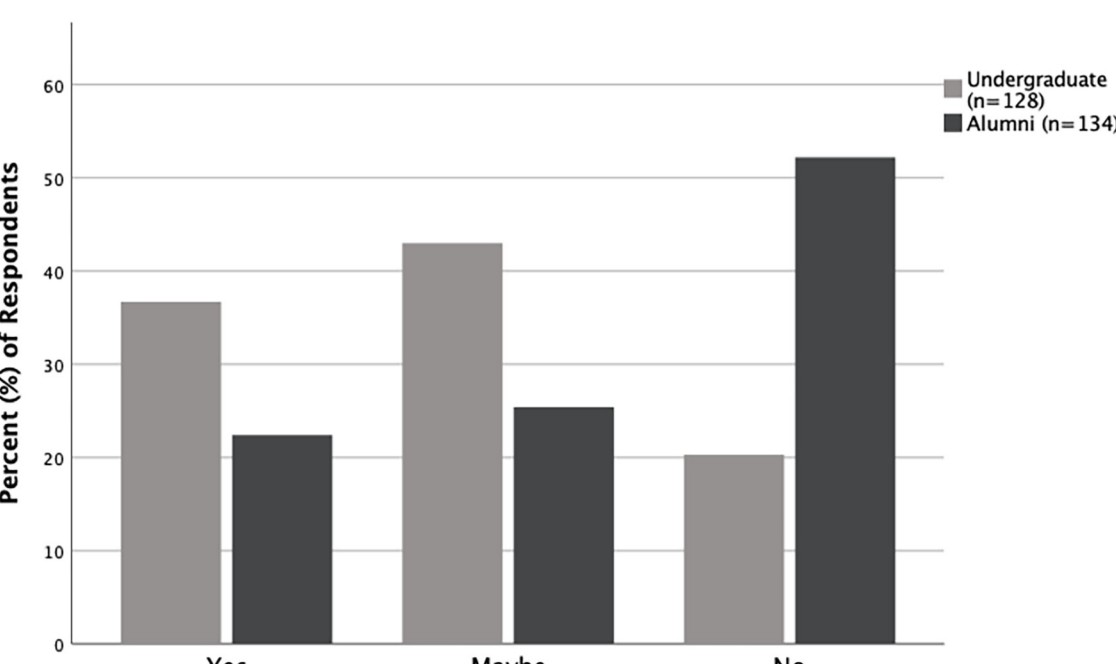

Did your undergraduate research experience prepare you for hostile work environments?

**Fig 4. Differences in alumni and undergraduates regarding preparation gained for hostile work environments.** Responses in percentages from alumni and undergraduates regarding if their undergraduate research prepared them for hostile work environments. Alumni and undergraduate responses were significantly different (p = 0.001).

responded that their undergraduate research experience did not prepare them for work opportunities due to their gender. For all other questions about their undergraduate research experience and mentor, there was no significant difference between the male and female respondents (S4 Table).

## Gender pairing of mentor-mentee

Questions that relate directly to the gender of the undergraduate research mentors were evaluated by comparing responses from those students who had male mentors to those that had female mentors; responses from both alumni and undergraduates were combined for this. There was a significant difference in how students and alumni with female mentors, compared to those with male mentors, responded to if the gender of their mentor contributed to their mentor-mentee relationship (p<0.001). A large majority (84.2%, n = 165) of students with male mentors, compared to only 58.1% (n = 124) of those with female mentors, responded that the gender of their mentor did not contribute to their relationship with their mentor (Fig 6A). Further, based on the gender of their mentor, there was also a difference in how students and alumni responded to the question of if their research experience had prepared them for a career in science. Only 76.5% (n = 183) of students with a male mentor compared to 87.4% (n = 135) of those with a female mentor, responded that their undergraduate research experience prepared them for a career in science (Fig 7). There was no difference in the responses from students with male mentors compared to those with female mentors when asked about other aspects relating to the gender of their mentor (S5 Table). We also analyzed the same set

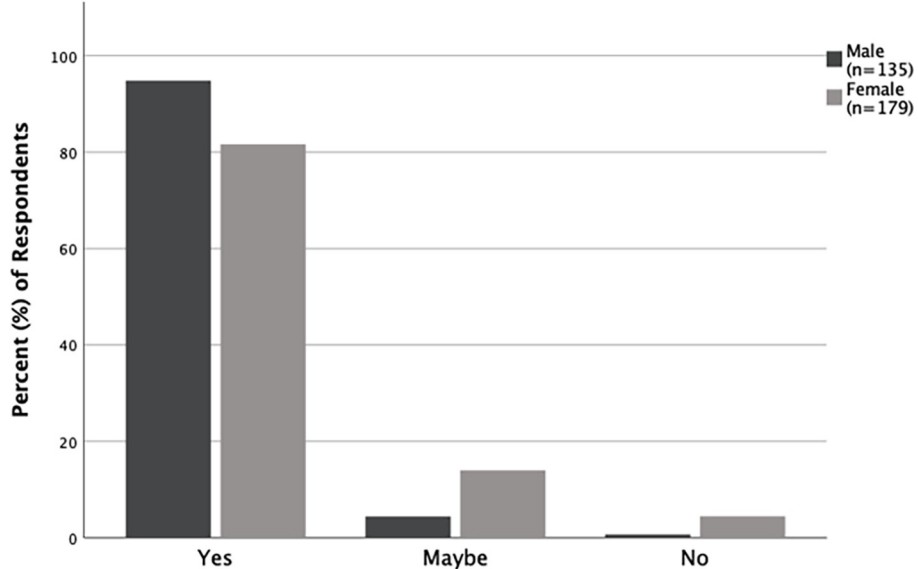

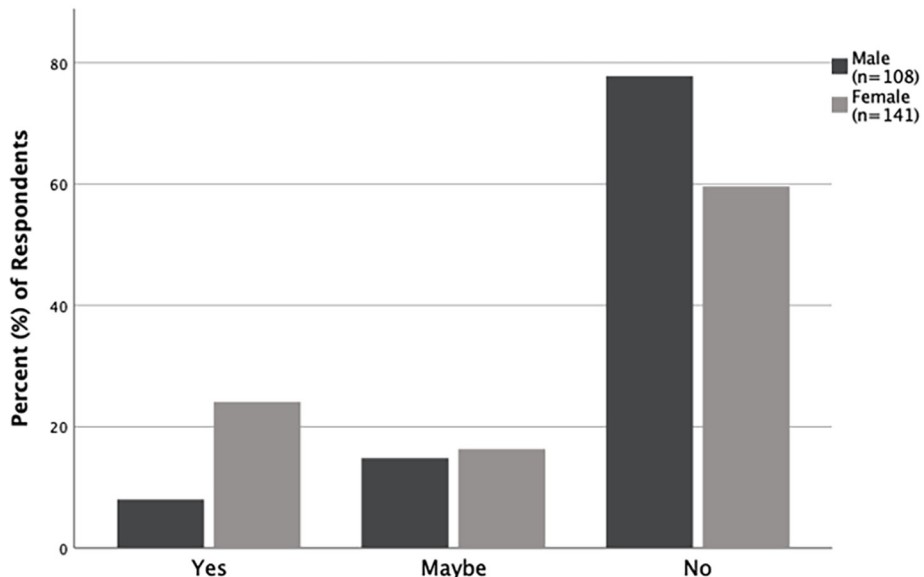

**Fig 5. Differences in male and female respondents regarding their views about undergraduate research and mentors.** Responses in percentages from female and male respondents (alumni and undergraduates) about (A) if they felt their research mentor was a good role model and (B) if they felt that their undergraduate research experience had prepared them for work opportunities due to their gender. Female and male respondents were significantly different (p = 0.001) for both questions.

of questions relating to the gender of the mentor by comparing how respondents (alumni and undergraduates combined) who were same gender-paired (i.e., as students, had mentors of the same gender as themselves) responded, compared to those who were different gender-paired

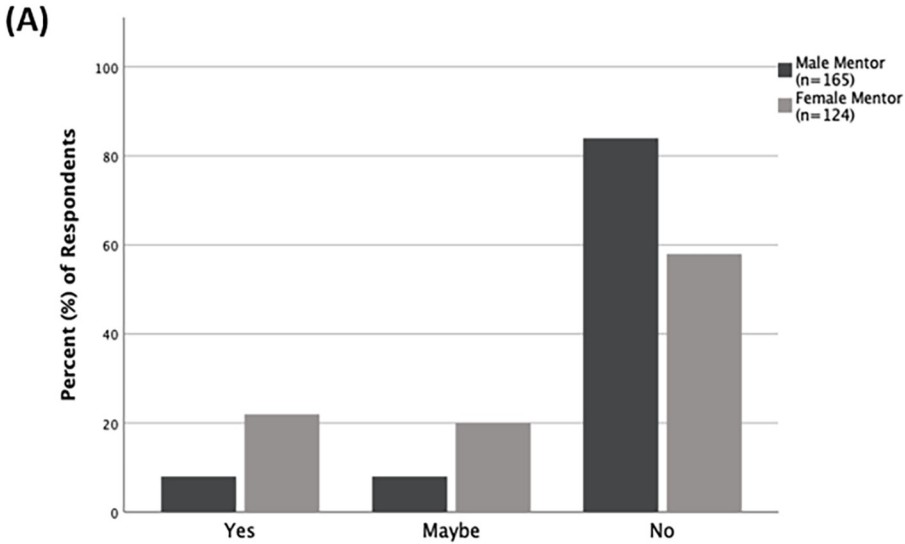

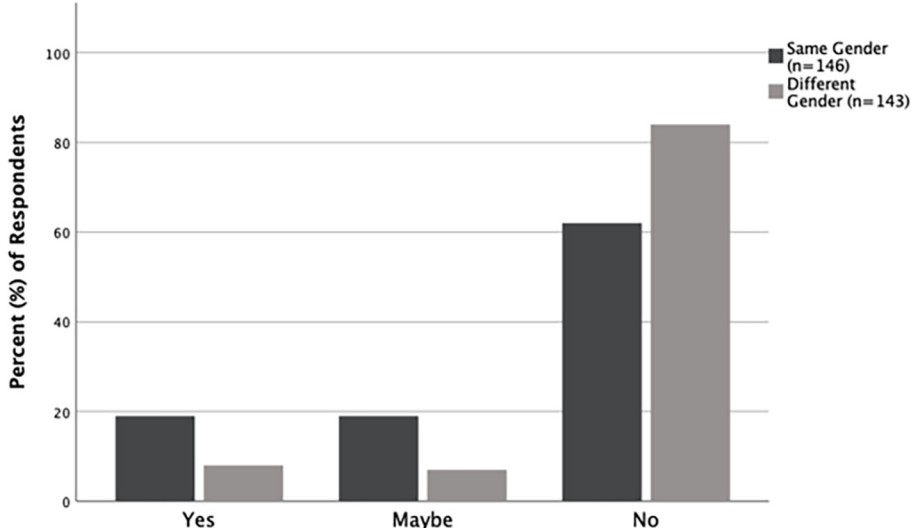

**Fig 6. Views about mentor gender and its contribution towards the mentor-mentee relationship.** Responses in percentages from alumni and undergraduates about if the gender of their mentor contributed towards their relationship with their mentor. Responses differed significantly (p<0.001) (A) between those who had male mentors and those who had female mentors, and (B) between those who were same gender-paired (mentor-mentee of same gender) and those who were different gender-paired (mentor-mentee of different genders).

(i.e., as students, had mentors of a different gender than themselves). Only 61.6% (n = 146) of same gender-paired students responded that the gender of their mentor did not contribute to their relationship, compared to 84.6% (n = 143) of different gender-paired students (p<0.001; Fig 6B). For all other questions about gender and mentoring, there was no difference based on gender-pairing (S5 Table).

To tease out if there was a difference between how different gender pairs responded to the question of if the gender of their mentor contributed to their mentor-mentee relationship, we also compared different gender pair combinations (i.e., male mentor-male student, male

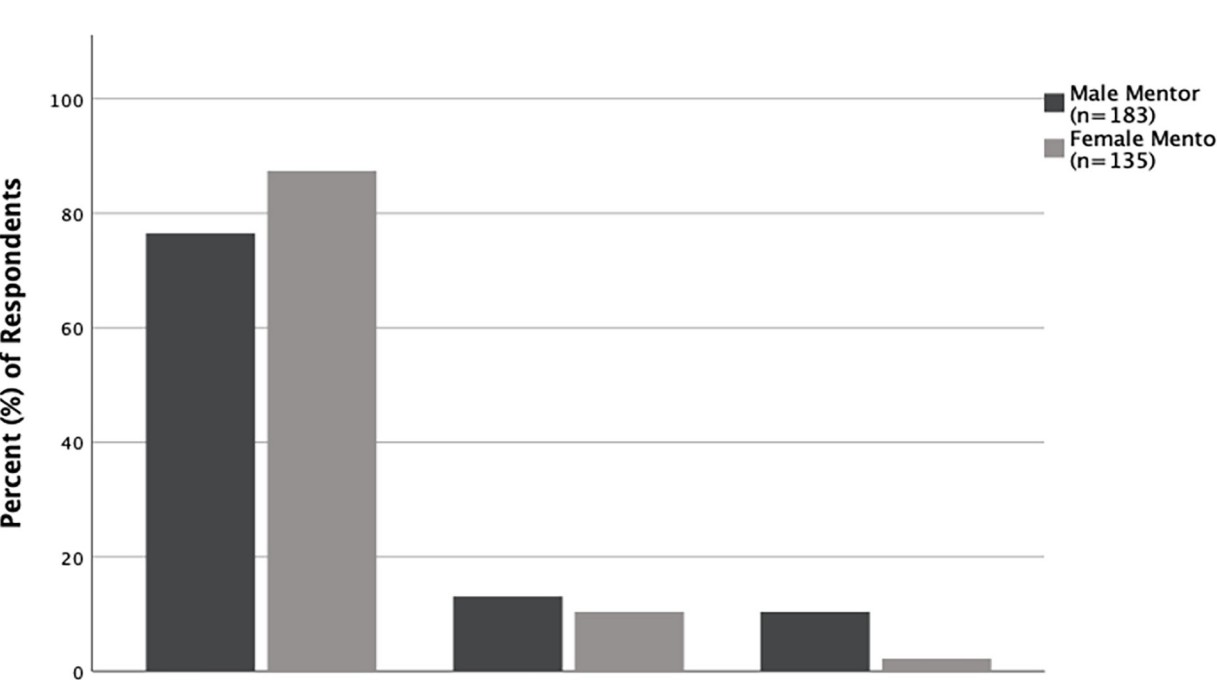

**Fig 7. Gender of mentor and how mentees responded about being prepared for a career in science.** Responses in percentages from alumni and undergraduates about if their undergraduate research experience prepared them for a career in science. Responses differed significantly (p = 0.008) between respondents who had male mentors and those that had female mentors.

mentor-female student, female mentor-male student, female mentor-female student). There was a significant difference between the responses from female students with female mentors and male students with male mentors (p<0.001). Less than half (46.1%, n = 76) of female students with female mentors, compared to 78.6% (n = 70) of male students with male mentors responded that the gender of their mentor did not contribute to their relationship with their mentor (Fig 8). Similarly, there was also a significant difference in how female students with female mentors responded, compared to how female students with male mentors responded. In this instance, 88.4% (n = 95) of female students with male mentors responded that the gender of their mentor did not contribute to their relationship with their mentor (Fig 8). Given that a smaller of portion of female students than males students responded that their mentor was a good role model (Fig 5A), we further investigated if the gender of their mentor had any influence on if female students perceived their mentor as a good role model. We found a significant difference (p = 0.005) in how female students responded based on the gender of their mentor. Almost ninety percent (89.9%, n = 79) of female students with female mentors stated that their mentor was a good role model, whereas only 76.7% (n = 102) of female students with male mentors stated so (Fig 9).

## Post-graduation experiences of alumni

Survey results showed that the majority (84.4%, n = 160) of alumni that responded were in a science-related field at the time of completing the survey. Over half (58.9%, n = 158) of the

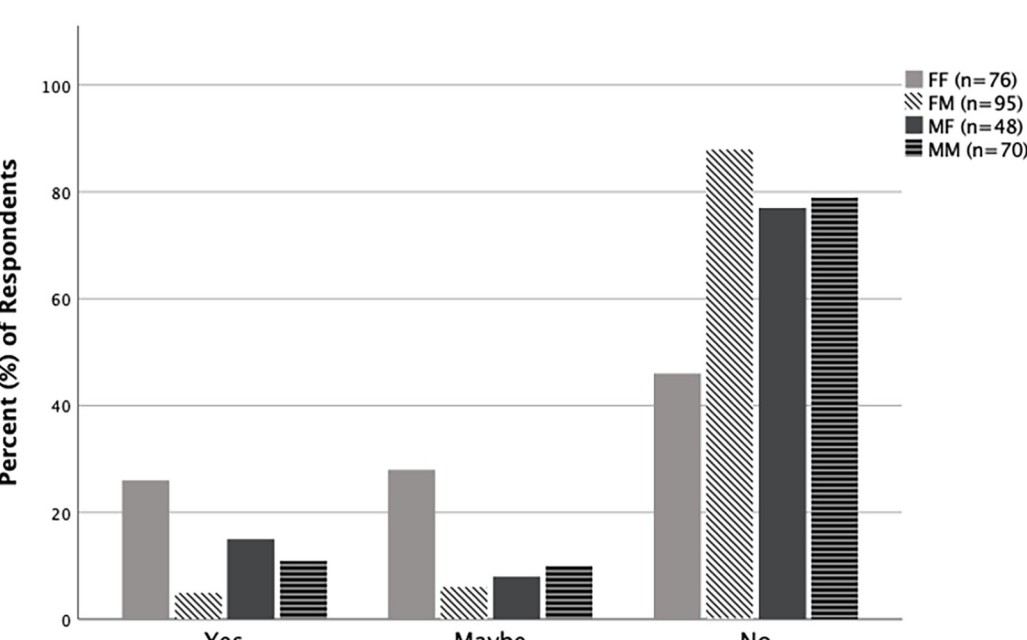

**Fig 8. Views on how gender of mentor contributed to the mentor-mentee relationship.** Responses in percentages from alumni and undergraduates in different mentor-mentee gender pairings. Responses differed significantly (p<0.001) between FF and FM, and between FF and MM. FF = Female students with female mentor, FM = Female students with male mentor, MF = Male students with female mentor, MM = Male students with male mentor.

alumni also responded that their undergraduate mentor and research experience had influenced their current education or employment. Further, 70.7% (n = 157) of responding alumni thought that their undergraduate research experience had adequately prepared them for their current education or employment status, and 83% (n = 153) responded that they thought the undergraduate research experience had adequately prepared female students for a career in science (Fig 10). There was no significant difference between how male and female alumni responded to these aspects of their undergraduate research experience (S6 Table). The majority of both male and female alumni who had a choice in selecting their current supervisor or mentor, responded that the gender of their undergraduate mentor did not influence the selection of their current mentor or supervisor (Fig 11A). However, when their responses were analyzed based on whether they had a choice in selecting their supervisor or mentor, a significantly higher proportion of female alumni compared to male alumni reported not having a choice in the selection of their current supervisor or mentor (p<0.01). Of the female alumni, 61.9% (n = 97) reported not having a choice when selecting their supervisor or mentor, while only 36.1% (n = 61) of male alumni reported not having a choice (Fig 11B).

## Discussion

Our data reveal that most alumni and undergraduate students surveyed feel that gender does not need to be considered when selecting a research mentor. In line with this, most of them reported that the gender of their undergraduate research mentor did not influence why they selected their mentor (Fig 1). We also did not see any association between student gender and

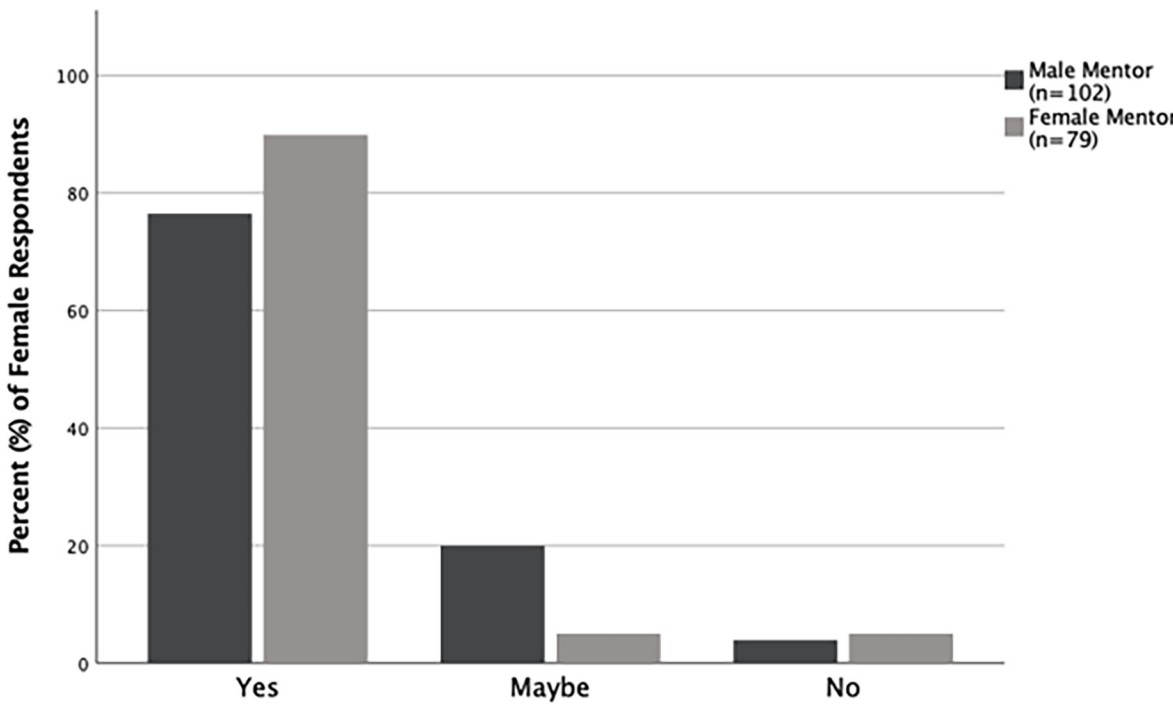

**Fig 9. Responses from female alumni and undergraduates regarding if they perceived their undergraduate mentor as a good role model.** Responses in percentages from female students who had a male mentor, and those who had a male mentor regarding if they thought their mentor was a good role model. Responses differed significantly (p = 0.005) between those who had male mentors and those who had female mentors.

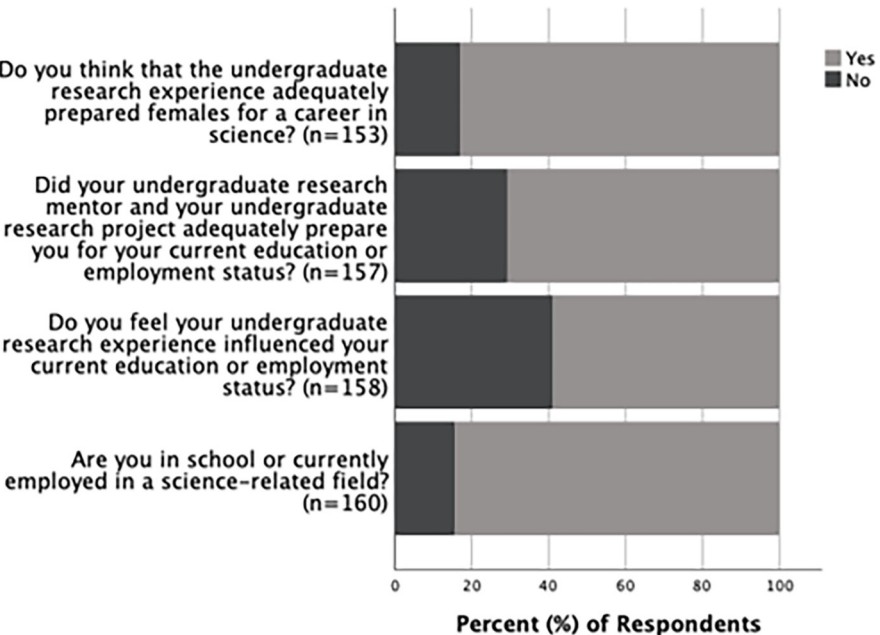

**Fig 10. Influence of undergraduate research on the current positions of alumni.** Responses in percentages from alumni regarding the influence of their undergraduate research experience on their current position.

Did gender influence the selection of your current mentor or supervisor

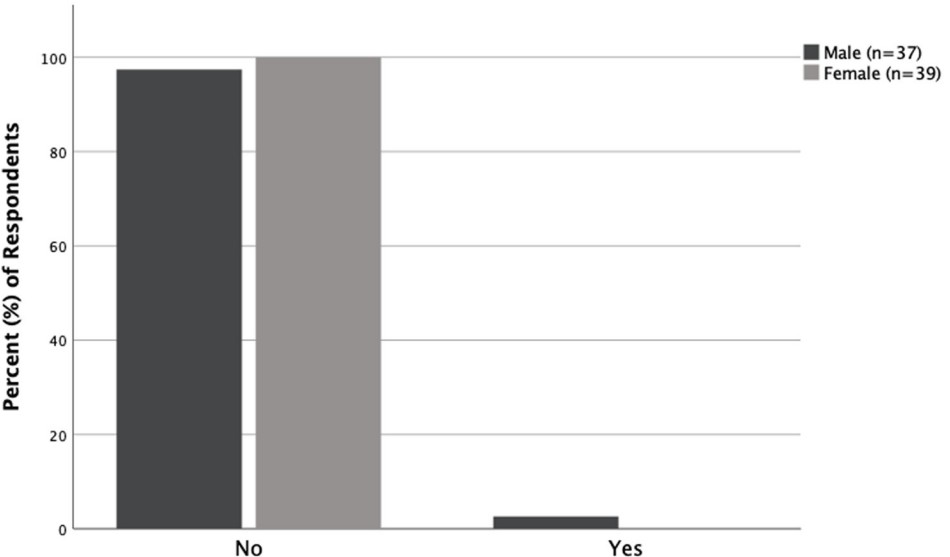

Choice in the selection of current mentor or supervisor

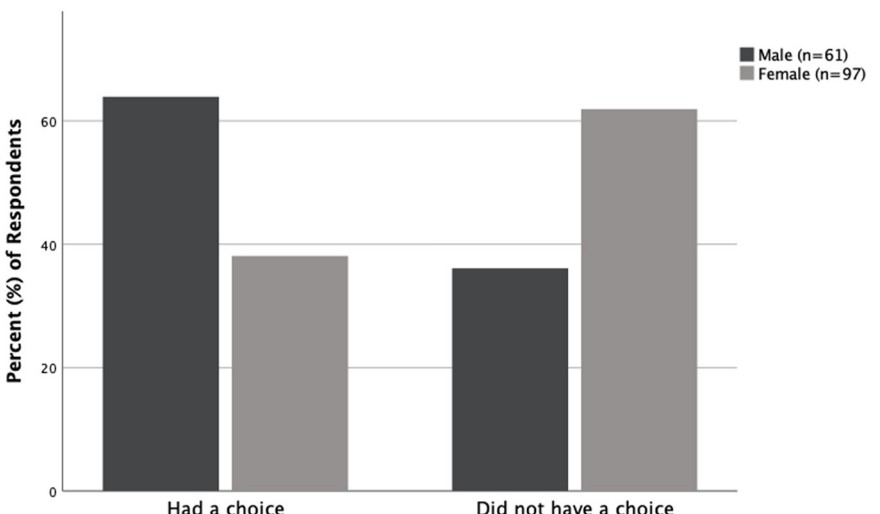

**Fig 11. Alumni choice in current mentor or supervisor selection.** (A) Male and female alumni responses in percentages regarding if gender of their undergraduate mentor played a role in the selection of their current mentor or supervisor, for those who had a choice. (B) Male and female alumni responses in percentages regarding if there was a choice when selecting the mentor or supervisor for their current position (p = 0.002).

mentor gender (S3 Table), indicating that mentor gender did not influence how students selected their mentor.

Although the majority of alumni and undergraduates said gender should not be considered when selecting a mentor, of those who responded that gender should be considered, there were more alumni than undergraduates (Fig 2A). It is possible that these differences are because alumni are likely to have had more work experience, including experience working

with mentors of different genders, than undergraduate students. Research suggests that female mentors provide more role modeling and less career development than male mentors [20]. Such differences in mentoring may have been observed by some alumni, leading them to state that gender needs to be taken into consideration when selecting a mentor. Further, the alumni surveyed had completed the program anytime between 1988 and 2016, therefore, a portion of them would have gone through an undergraduate experience when the UNK Biology department faculty was largely composed of men. This imbalance in gender representation of mentors may have skewed notions and expectations about male and female mentors and may also be a contributing factor to why alumni may have considered the gender of their mentor as important. For future studies, it will be valuable to collect data on the specific years that alumni participated in the program to assess how gender imbalances in faculty composition relate to student perceptions of mentor selection.

Surprisingly, when asked if female mentors should mentor male students and male mentors should mentor female students, the responses from alumni and undergraduates were mixed (Fig 1). These mixed responses contradict their overall consensus about gender not being necessary to consider when selecting a mentor. Therefore, there is a possibility that this question was interpreted in different ways by the respondents. Some respondents may have misinterpreted the question as asking if only female mentors should mentor male students and if only male mentors should mentor female students. Since there is uncertainty about how this question was interpreted, even though we observed a difference in how undergraduates and alumni answered this question (Fig 2B), it is not currently possible to interpret this particular result unambiguously. Future investigations with more clarity regarding this question will be required to determine perceptions towards mentor-mentee relationships comprising of different genders (i.e., male mentors for female students and female mentors for male students).

Overall, responses from undergraduates and alumni demonstrated that their undergraduate research experience was positive. The Biology department mentors at UNK were also regarded as being good role models, helpful, understanding, and available to help by both undergraduates and alumni (Fig 3). Our results agree with past studies that have also found similar benefits of undergraduate science research [2, 4, 5]. Responses from alumni and undergraduates about their undergraduate research experience were similar, except regarding if their undergraduate research experience had prepared them for hostile work environments. The alumni felt less prepared for hostile work environments compared to undergraduates (Fig 4). One of the reasons for this difference could be attributed to some of the alumni not completing the Biology research series at UNK with hostile work environment training, because this training was not implemented until 2010. Further, some alumni may have also only experienced undergraduate research in a time when most research mentors in the department were men and may not have considered work-related hostility, especially as it relates to women. Another reason for the difference may be because of the differences in their years of work experience. Alumni are likely to have had more years of work experience compared to undergraduates and, therefore, may have a better perspective of hostile work environments and what types of preparations help with navigating such situations. While it is positive that most undergraduates felt prepared for hostile work environments, it is important to consider that more than half of the alumni did not. This emphasizes the continued need to prepare students for hostile work environments that continue to exist in STEM fields [21]. Further investigation, specifically of alumni who had undergone the hostile work environment training as students, will be useful to determine if the undergraduate research program accompanied by the hostile work environment training was useful in preparing students for hostile work environments.

A comparison of responses from male and female students shows that both genders shared similar perceptions about their undergraduate research experience and research mentor.

However, there were key differences between the two genders, namely, if they perceived their mentor as a good role model, and if their undergraduate research experience had prepared them for work opportunities due to their gender. Even though most female and male respondents thought their mentor was a good role model, a smaller proportion of female respondents compared to male respondents, responded as such (Fig 5A). Our analysis indicates that how female students perceive their mentor may be related to their mentor's gender since, female students with female mentors were more likely than female students with male mentors to report that their mentor was a good role model (Fig 9). These data reiterate past studies that demonstrate the value of female role models for female trainees [15, 16, 22].

Most of both male and female students responded that their undergraduate research experience did not prepare them for work opportunities due to their gender (Fig 5B). Since most respondents felt underprepared for work opportunities due to their gender, it suggests that overall, they do not necessarily feel that work opportunities are specific to particular genders. However, female students were more likely than male students to state that their undergraduate research experience had prepared them for opportunities due to their gender. This difference indicates that women, more than men, may associate work opportunities with their gender, which may be a consequence of gender stereotypes that women are often exposed to starting from an early age [15, 23].

Most alumni and undergraduate students responded that the gender of their undergraduate mentor did not influence their relationship with their mentor. However, a larger proportion of students with female mentors than those with male mentors, stated that the gender of their mentor had or may have influenced their relationship with their mentor (Fig 6A). Similarly, students with a mentor of the same gender as themselves were more likely to state that gender of their mentor had or may have influenced their relationship with their mentor, compared to those with a mentor of a different gender than themselves (Fig 6B). Female students with female mentors were most likely than any other mentor-mentee gender-pair to state that their mentor's gender influenced their relationship with their mentor (Fig 8). This is likely because women often feel more comfortable working with other women, and female STEM professors appear to act as positive role models in science, as well as reduce the cultural stereotype that science is a male-dominated field [22, 24]. Interestingly, regardless of their own gender, when respondents stated that the gender of their mentor contributed to their relationship with their mentor, their mentor was more likely to be female. Another area where the gender of the mentor influenced responses was regarding if students felt that their undergraduate research experience had prepared them for a career in science. While most alumni and undergraduates felt they had received preparation, a higher proportion of respondents who had female mentors responded so (Fig 7). These findings together support the utility of female mentors for undergraduate students in general, regardless of the gender of the students.

Responses from the alumni regarding the influence of undergraduate research on their current positions demonstrated that for the majority of graduates, the undergraduate research experience had a positive influence on their current position. Moreover, most of the graduates were in a science-related field, indicating that the graduates were successful in pursuing science-related careers, post-graduation (Fig 10). However, female alumni were more likely than male alumni to be in a situation where they did not have a choice in selecting their current mentor or supervisor, regardless of if they wanted to select their mentor or supervisor based on gender (Fig 11B). These data indicate that female graduates may be experiencing the workplace in a different way compared to their male counterparts. The lack of choice that female alumni reported could be due to fewer job opportunities and therefore fewer mentors being available for them. It may also be because men are more likely to be in senior positions (such as supervisors or mentors) than women are in STEM fields [12]. Consequently, female alumni

may have been presented with only one gender (male) as a possible mentor or supervisor, leaving them no choice for selecting one of their chosen gender.

While our study provides valuable insights on undergraduate research mentoring relationships, it is important to note that the results of our study are based on a single department within our institution. Further, our respondents represent a population that is predominantly white. Therefore, this may limit how generalizable our findings are. Larger studies across multiple institutions with a diverse sample will be necessary in the future. Further, the scope of our study is restricted to binary gender identities, and studies that take into account non-binary gender identities will be needed going forward. Another factor to consider for future work would be the possible heterogeneity (other than gender) in the research mentors. To account for this, we suggest future work will need to look at what specific attributes of mentors, irrespective of their gender, may influence the research experience of students. Finally, since we surveyed alumni who graduated at different times over a span of almost 30 years, there is a possibility that there were varying levels of recall bias amongst the alumni when answering some questions about their undergraduate research experience. However, recall bias would not have been an issue for questions pertaining to the alumni's perspective about mentor selection and about their current position. Even with these limitations, we believe our findings still provide valuable insights regarding the role that gender plays in mentoring relationships and are applicable to institutions with an emphasis on undergraduate research. In particular, the components of our program that likely contribute to its success include, the hostile work environment training, and the 2-part structure of the program in which students get to know their mentors while designing their study in the first part, before they carry out their research project in the second part. These features likely helped with preparing students for hostile work environments and with allowing students to form constructive mentor-mentee relationships. We suggest that these features would be possible and worthwhile to replicate in other institutions to enhance the undergraduate research experience for students.

## Conclusions

Our study shows that undergraduate students do not display any conscious biases when selecting mentors, but they seem to perceive mentors differently based on their gender. Specifically, female mentors were more likely to be regarded by both male and female students as having a positive influence, than male mentors in terms of preparing students for a career in science. Students that reported that the gender of their mentor contributed to the mentor-mentee relationship were more likely to have a female mentor. Further, female students with a female mentor were more likely than those with a male mentor to state that their mentor was a good role model. While the benefit of female mentors for female students that we report here is expected and previously documented [16, 22], our study shows that female mentors were beneficial for male students as well. These results add to our current knowledge regarding undergraduate mentoring relationships and underscore the value of female mentors in undergraduate research. Therefore, for institutions with undergraduate research programs, it is important to consider that female representation in research and mentoring roles can be useful for all undergraduate students. Male academics should also be encouraged and trained to be effective and inclusive mentors. Crucially, it will be useful for men and women to be equally represented in research and mentoring roles in STEM departments.

Overall, the undergraduate research experience at UNK was regarded as useful by both undergraduate students and alumni. In particular, the fact that most alumni not only thought of their research experience as positive but also were working or studying further in science-related fields, is a testament to the success of the undergraduate research program at UNK.

Our undergraduate research program appears to be performing its intended purpose of preparing students for careers in science and our study supports the continued implementation of undergraduate research programs for the benefit of students. Importantly, the findings of our study are helpful for the broader STEM community to understand what type of research mentoring relationships can benefit current and future students. A deeper understanding of what features of the undergraduate research experience enable students to succeed in STEM careers is still needed and will assist in strengthening undergraduate mentoring and research programs.

## Supporting information

**S1 Table. Summary of sample and response rates.**
(PDF)

**S2 Table. Summary of responses to gender and mentor selection questions.**
(PDF)

**S3 Table. Summary of assignment of mentors based on student gender and mentor gender.**
(PDF)

**S4 Table. Summary of responses to questions about undergraduate research experience and mentors.**
(PDF)

**S5 Table. Summary of responses to questions about gender and research mentor.**
(PDF)

**S6 Table. Summary of alumni responses to questions about undergraduate research experience and current positions.**
(PDF)

**S1 Appendix. Undergraduate research mentoring survey.**
(PDF)

**S1 File. Excel dataset containing survey responses.**
(XLSX)

## Acknowledgments

We would like to thank Jyoti Iyer for critical feedback on this manuscript and Hari K. Iyer for consultation on statistical analysis.

## Author Contributions

**Conceptualization:** Julie J. Shaffer, Kimberly A. Carlson.

**Data curation:** Katelyn Baumgart.

**Formal analysis:** Saili Moghe.

**Investigation:** Julie J. Shaffer, Kimberly A. Carlson.

**Methodology:** Julie J. Shaffer, Kimberly A. Carlson.

**Supervision:** Julie J. Shaffer, Kimberly A. Carlson.

**Visualization:** Saili Moghe.

**Writing – original draft:** Saili Moghe, Katelyn Baumgart.

**Writing – review & editing:** Saili Moghe.

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
