## [Decision Letter · Decision Letter 0]

26 Aug 2021

PONE-D-21-21231

The Importance of female mentoring in undergraduate STEM research experiences

PLOS ONE

Dear Dr. Moghe,

Thank you for submitting your manuscript to PLOS ONE. My apologies for taking so long, due to the uncommon circumstances. After careful consideration, we feel that it has merit but does not fully meet PLOS ONE’s publication criteria as it currently stands. Therefore, we invite you to submit a revised version of the manuscript that addresses the points raised during the review process.

I have received two reviews of your manuscript. Both reviewers think the topic of the study is relevant. However, they also encountered several problems in your manuscript, which are detailed below. I also identified a number of issues based on my independent review. The consistent problem we all found can be solved by describing, under the limitations of this study, the potential response bias due to social desirability. 

A rebuttal letter that responds to each point raised by the reviewers. You should upload this letter as a separate file labeled 'Response to Reviewers'.A marked-up copy of your manuscript that highlights changes made to the original version. You should upload this as a separate file labeled 'Revised Manuscript with Track Changes'.An unmarked version of your revised paper without tracked changes. You should upload this as a separate file labeled 'Manuscript'.

We look forward to receiving your revised manuscript.

Kind regards,

Juan Cristobal Castro-Alonso, Ph.D.

Academic Editor

PLOS ONE

Journal Requirements:

2. Please consider changing the title so as to meet our title format requirement (https://journals.plos.org/plosone/s/submission-guidelines). In particular, the title should be "Specific, descriptive, concise, and comprehensible to readers outside the field" and in this case it is not informative and specific about your study's scope and methodology.

3. Please change "female” or "male" to "woman” or "man" as appropriate, when used as a noun (see for instance https://apastyle.apa.org/style-grammar-guidelines/bias-free-language/gender)

Reviewers' comments:

Reviewer's Responses to Questions

**Comments to the Author**

1. Is the manuscript technically sound, and do the data support the conclusions?

Reviewer #1: Partly

Reviewer #2: Partly

2. Has the statistical analysis been performed appropriately and rigorously? 

Reviewer #1: Yes

Reviewer #2: Yes

3. Have the authors made all data underlying the findings in their manuscript fully available?

Reviewer #1: Yes

Reviewer #2: Yes

4. Is the manuscript presented in an intelligible fashion and written in standard English?

Reviewer #1: Yes

Reviewer #2: Yes

5. Review Comments to the Author

Reviewer #1: Thank you for the opportunity to review this interesting article. I believe this is a well-written piece of research with objectives that are well justified by a good literature review. However, the study has some important limitations, that the authors fail to identify and discuss appropriately. I, therefore, recommend accepting this manuscript once the authors have addressed the following issues:

- Key limitations of the study are social desirability and recall bias in responses. Please discuss these limitations.

- Limits to generalizability of findings should also be discussed as the study was based on the case of one program at one institution.

- The underrepresentation of women in STEM fields is mentioned in at least three different parts of the introduction, which makes the text a bit repetitive. I believe it is better to discuss this once, thoroughly, in the introduction.

- In my opinion, the following paragraph on page 3 is tautological: “Although the percentage of women working in science-related careers has improved greatly, women are still underrepresented in these fields [13]. Men have also been more widely acclaimed in science fields than women. Women represent approximately 48% of the United States work force, but they only represent 27% of STEM careers [13]. This is an improvement from 8% of women in STEM careers in 1970 but is still disproportional [13]. This gender gap may be the reason why women are more likely to work in education or healthcare careers, rather than STEM careers [16]”. Perhaps it would be better to mention sociological explanations for horizontal gender segregation in academic fields.

- This paragraph on pages 3-4 is somewhat repetitive: “The purpose of our study was to learn how effective the UNK undergraduate research program has been in enabling students to succeed in the sciences, post-graduation. Additionally, our study aimed to determine if there were any differences in the research experience, including mentoring relationships, of students based on their own gender and their mentor’s gender. Here we report our findings on the effectiveness of the undergraduate research program, and the differences in the undergraduate research experience of students based on gender.”

- In my opinion, the subsection “Undergraduate students vs. alumni – mentor selection” in the results section does not contribute to addressing the study’s objectives. It has more to do with the data and processes of mentor selection. Therefore, I would suggest including this information in the Methods sections. Further, a key issue would be to actually test if assignment to male and female mentors differed by student gender. This is, Fisher’s exact test could also be used to assess the association between student gender and mentor gender.

- Reliability issues: Some of the items were worded in ways that could have been confusing for the respondents and, therefore, interpretating these results can be challenging. For example, whether the mentorship “prepare them for work opportunities due to their gender” or the statement “females should mentor males and vice versa”. I would suggest not reporting these results.

- When discussing post-graduation experiences, the authors refer to alumni’s current mentors. Is this a figure that they all should encounter in their careers? How common is to have a mentor? What did participants understand for this (e.g. employer, supervisor)?

- In the discussion section, the authors suggest that the year in which students completed the program could have an impact on their perceptions, due to changes in the imbalances in gender representation of mentors. I would suggest testing this and reporting the results in the paper.

- On page 12, the authors state “Why a greater proportion of females than males may not have perceived their mentor as a good role model warrants further investigation.” Could they test if females that had males as mentors were more likely to perceive this?

- Finally, on page 13, the authors state: “Therefore, it is important to consider that female representation in mentoring roles can be useful for undergraduate students of all genders, not just females. Our results show that female mentors benefit both male and female undergraduate trainees and are also in agreement with past findings which show that female mentors benefit female trainees [22].” However, readers could interpret this as a recommendation to assign women, more frequently than men, to mentoring roles, and I find this problematic. In academia, women usually tend to be assigned to teaching, supervision and administrative duties more frequently than men, and this can have an impact on their careers as they are left with less time for other areas that are highly valued for promotion, such as research. Recommending disproportionally position female academics in mentoring roles could increase this burden, particularly in context with low representation of women. In my opinion, male academics should also be encouraged and trained to be to be effective and inclusive mentors.

Reviewer #2: Overall

This paper evaluates how the gender of undergraduate research mentors influences the research experience of students. I agree with the authors that this is a relevant topic, that needs to be investigated further. However, I have some concerns with this paper as it is currently written. Particularly, with the methods and analysis sections.

Methods & Analysis:

1. Description of the overall sample and response rate would be helpful. Particularly, it would be useful to understand the different response rates among groups: gender, students/alumni, cohorts, and majors. Are these response rates consistent with the UNK undergraduate student population?

2. Potential response bias is a key component to this study. However, authors don’t mention nor elaborate on this. How does response bias play a role in the analysis? How is this study addressing this issue? Are these results robust?

3. Related to the above, I wonder how students/alumni academic outcomes and demographics are related to response rates and survey answers. Are there subgroups of the UNK population that are heavily represented in the survey results?

4. Authors mention the use of Fishers Exact Test to answer research questions. Elaborate on why you choose this method over Chi-square test, specially giving your sample size.

5. How was the survey constructed? Was there any psychometric analysis regarding validity or reliability of the measure? How is this related to different interpretations to some items by the respondents?

6. The authors mentioned how future studies should be directed towards determining the attributes of mentors that make them good role models. On this note, was there any attempt to account for mentor heterogeneity? (other than gender). If not, how can attributes of mentors influence these results? Is this a potential limitation to this study?

Conclusion & Discussion:

7. It appears that the undergraduate research experience at UNK was viewed as useful -according to authors analyses. How can these results be of any help to other institutions? What are the main elements of this research experience that added value? Are they any different to other mentoring programs in STEM? Can or should this be replicated in other institutions? If so, what are the authors recommendations.

8. It would be important to address the limitations of the study. Are these results generalizable to other contexts or institutions? Or even to other departments within UNK?

9. I would suggest further developing ideas in the conclusion section. How are these results pertinent to this specific journal audience? What did the authors learn about phenomenon through analyzing data that can contribute to the existing literature?

6. PLOS authors have the option to publish the peer review history of their article (what does this mean?). If published, this will include your full peer review and any attached files.

Reviewer #1: No

Reviewer #2: No

---

## [Author Response · Author response to Decision Letter 0]

18 Oct 2021

Dear Reviewers,

We thank you for the helpful and thorough review of our manuscript. We are grateful and appreciative of the time and effort spent on the review of our manuscript. The critical comments that we have obtained from you have been valuable in greatly improving our manuscript. 

As you will note from our revised manuscript, we have spent a considerable amount of effort in addressing every editorial and reviewer comment that we received, as thoroughly as possible. In this letter we have responded to the journal requirements and each point raised by Reviewer # 1 and Reviewer # 2. The line numbers that have been referenced in our responses correspond to the unmarked version of our revised manuscript (the file ‘Revised Manuscript’).

Journal Requirements:

We have ensured that our manuscript meets the PLOS ONE style requirements, including those for file naming. The manuscript has been formatted in accordance with the following PLOS ONE style templates: (https://journals.plos.org/plosone/s/file?id=wjVg/PLOSOne_formatting_sample_main_body.pdf and 

https://journals.plos.org/plosone/s/file?id=ba62/PLOSOne_formatting_sample_title_authors_affiliations.pdf). 

2. Please consider changing the title so as to meet our title format requirement (https://journals.plos.org/plosone/s/submission-guidelines). In particular, the title should be "Specific, descriptive, concise, and comprehensible to readers outside the field" and in this case it is not informative and specific about your study's scope and methodology.

We have now reworded our title so that it is informative and specific about the scope and methodology of our study. The title has been changed from “The importance of female mentoring in undergraduate STEM research experiences” to “Female mentors positively contribute to undergraduate STEM research experiences.”

3. Please change "female” or "male" to "woman” or "man" as appropriate, when used as a noun (see for instance https://apastyle.apa.org/style-grammar-guidelines/bias-free-language/gender)

We have now changed “female” to “woman” and “male” to “man” when these have been used as a noun in the entire revised manuscript. 

Response to comments from Reviewer # 1

We would like to thank reviewer #1 for their thorough review of our manuscript and their valuable suggestions and feedback provided to improve our manuscript. Below we have addressed each of the comments from Reviewer #1.

- Key limitations of the study are social desirability and recall bias in responses. Please discuss these limitations.

We have now included in our methods section the steps that were taken to reduce social desirability and recall bias in survey responses. To limit social desirability, the survey was completely anonymous with no identifiers. In addition, the survey was worded such that the respondents could not be identified in any way and so that the questions would not be leading. To minimize recall bias, the survey was given to the undergraduates during the semester that they were enrolled in the research series (lines 90-94). There is a possibility of recall bias in alumni responses given that they may have completed the research series at any time within the timespan of 1988-2016, and we have discussed this limitation in the discussion section (lines 367-370). 

- Limits to generalizability of findings should also be discussed as the study was based on the case of one program at one institution.

We have now included in our discussion section a portion that discusses the extent of the generalizability of our findings. In our discussion we acknowledge that our study is based on data from a single program within a single institute. We explain how our findings can be applicable to other institutes that have the same emphasis on undergraduate research as our institute (lines 359-378). 

- The underrepresentation of women in STEM fields is mentioned in at least three different parts of the introduction, which makes the text a bit repetitive. I believe it is better to discuss this once, thoroughly, in the introduction.

We have now consolidated our discussion regarding the underrepresentation of women in STEM fields into one comprehensive paragraph (lines 43-52). 

- In my opinion, the following paragraph on page 3 is tautological: “Although the percentage of women working in science-related careers has improved greatly, women are still underrepresented in these fields [13]. Men have also been more widely acclaimed in science fields than women. Women represent approximately 48% of the United States work force, but they only represent 27% of STEM careers [13]. This is an improvement from 8% of women in STEM careers in 1970 but is still disproportional [13]. This gender gap may be the reason why women are more likely to work in education or healthcare careers, rather than STEM careers [16]”. Perhaps it would be better to mention sociological explanations for horizontal gender segregation in academic fields.

We have reworded this section to reduce any redundancies; the relevant information is now in the consolidated paragraph about the underrepresentation of women in STEM fields (the same paragraph mentioned in response to the previous reviewer comment, lines 43-52). As suggested, we have also added information about sociological explanations for horizontal gender segregation in academic fields (lines 49-52).

- This paragraph on pages 3-4 is somewhat repetitive: “The purpose of our study was to learn how effective the UNK undergraduate research program has been in enabling students to succeed in the sciences, post-graduation. Additionally, our study aimed to determine if there were any differences in the research experience, including mentoring relationships, of students based on their own gender and their mentor’s gender. Here we report our findings on the effectiveness of the undergraduate research program, and the differences in the undergraduate research experience of students based on gender.”

We have now removed any repetitive statements from this paragraph (lines 60-68) to make it concise.

- In my opinion, the subsection “Undergraduate students vs. alumni – mentor selection” in the results section does not contribute to addressing the study’s objectives. It has more to do with the data and processes of mentor selection. Therefore, I would suggest including this information in the Methods sections. Further, a key issue would be to actually test if assignment to male and female mentors differed by student gender. This is, Fisher’s exact test could also be used to assess the association between student gender and mentor gender.

As suggested, we have carried out the Fisher’s exact test to assess if there is an association between student gender and mentor gender. Our analysis shows that there is no significant difference in the assignment to male and female mentors based on student gender. We have reported these results in our revised paper, and they are stated in a newly created section entitled, “Mentor selection and mentor assignment” (lines 141-142). In this section we have also retained the results regarding mentor selection (lines 123-127) because we feel that the data about mentor selection still contributes to our study objectives of understanding how gender influences the mentor-mentee relationship. Since the mentor-mentee relationship first starts with mentor selection, we feel that reporting the data on mentor selection provides insights about how the mentor-mentee relationship is initiated and therefore have kept these results in the results section of our revised manuscript.

- Reliability issues: Some of the items were worded in ways that could have been confusing for the respondents and, therefore, interpretating these results can be challenging. For example, whether the mentorship “prepare them for work opportunities due to their gender” or the statement “females should mentor males and vice versa”. I would suggest not reporting these results.

We agree that the data for some of the questions are challenging to interpret given that some questions may have been interpreted by respondents in different ways. To this end, in the discussion section, we have explained that some of these questions may have been confusing to respondents and have also detailed how these questions may have been misinterpreted (lines 286-293). While we agree that not reporting the results of these questions will remove any confusion with data interpretation, we feel that reporting the data provides us the avenue to discuss the need to still address such questions and better word these questions in future studies (lines 291-295).

- When discussing post-graduation experiences, the authors refer to alumni’s current mentors. Is this a figure that they all should encounter in their careers? How common is to have a mentor? What did participants understand for this (e.g. employer, supervisor)?

When discussing post-graduation experiences, we refer to the alumni’s current mentor based on the survey questions that are about the current supervisor or mentor of the alumni. Our survey questions for these were phrased as, “What is the gender of your current supervisor or mentor?” and “Did the gender of your undergraduate research mentor influence your choice of your current supervisor or mentor?” Therefore, based on the wording of the question, alumni would have regarded this question to be referring to either their mentor or supervisor. These questions are based on the expectation that post-graduate careers or education typically involve having a mentor or supervisor. In our reporting and discussion of the data (lines 252-258; lines 350-358) and in the associated figure (Fig 11) and figure legend (lines 262-265) displaying data of alumni mentors and supervisors, we have now made sure to specify that we are referring to both mentors and supervisors and not just mentors. We thank the reviewer for bringing this to our attention so that we could edit this appropriately in our revised manuscript.

- In the discussion section, the authors suggest that the year in which students completed the program could have an impact on their perceptions, due to changes in the imbalances in gender representation of mentors. I would suggest testing this and reporting the results in the paper.

Our survey was open to students who were in the program between 1988 and 2016. However, since our survey did not collect information regarding which year students graduated or completed the program, we do not have the data to be able to test if the year in which students completed the program influenced their perceptions. Our survey was limited to asking if students had completed the program up to 6 or more years ago but not specifically how much more than 6 years ago. Therefore, we do not have the data to test the association between student perceptions and the faculty gender ratio at the time they were in the program. Our discussion proposes the possibility that the year(s) in which students participated in the research courses could influence their perceptions, due to changes in the imbalances in gender representation of mentors throughout the history of the department. We have further clarified in our discussion that this hypothesis should be tested in future studies which should collect data regarding what years students were in the program. Collection of such data can then be used to determine if there is an association between student perceptions and the ratio of male to female faculty mentors during the years in which students were in the program (lines 279-285).

- On page 12, the authors state “Why a greater proportion of females than males may not have perceived their mentor as a good role model warrants further investigation.” Could they test if females that had males as mentors were more likely to perceive this?

We have now looked at how female students with male mentors and female students with female mentors responded to the question of if they thought their mentor was a good role model. Based on our analysis, there was a significant difference in how female students responded based on the gender of their mentor. We have reported these results (lines 227-232) in the revised paper and displayed the results in Fig 9. We have also discussed this finding in our discussion (lines 320-323). We appreciate the reviewer’s insight and for directing us to perform this analysis.

- Finally, on page 13, the authors state: “Therefore, it is important to consider that female representation in mentoring roles can be useful for undergraduate students of all genders, not just females. Our results show that female mentors benefit both male and female undergraduate trainees and are also in agreement with past findings which show that female mentors benefit female trainees [22].” However, readers could interpret this as a recommendation to assign women, more frequently than men, to mentoring roles, and I find this problematic. In academia, women usually tend to be assigned to teaching, supervision and administrative duties more frequently than men, and this can have an impact on their careers as they are left with less time for other areas that are highly valued for promotion, such as research. Recommending disproportionally position female academics in mentoring roles could increase this burden, particularly in context with low representation of women. In my opinion, male academics should also be encouraged and trained to be to be effective and inclusive mentors.

We have now reworded this portion to clarify for the readers that we are not at all recommending that women should be assigned to mentoring roles more frequently than men, but rather, that women and men should be equally represented in research and in research mentoring roles (386-393). We also agree with the reviewer’s opinion that male academics should be encouraged and trained to be effective and inclusive mentors, and therefore, we have also added this apt recommendation within our manuscript (lines 391-392). We thank the reviewer for this suggestion and for pointing out that our original wording could have been misinterpreted.

Response to comments from review # 2

We would like to thank reviewer #2 for their thorough review of our manuscript and their valuable suggestions and feedback provided to improve our manuscript. Below we have addressed each of the comments from reviewer #2.

1. Description of the overall sample and response rate would be helpful. Particularly, it would be useful to understand the different response rates among groups: gender, students/alumni, cohorts, and majors. Are these response rates consistent with the UNK undergraduate student population?

We have now described the overall sample and explained that the sample is reflective of the UNK undergraduate population. We have also included the response rates of undergraduates and alumni (lines 107-115).

2. Potential response bias is a key component to this study. However, authors don’t mention nor elaborate on this. How does response bias play a role in the analysis? How is this study addressing this issue? Are these results robust?

We have now included in our methods section the steps that were taken to reduce social desirability and recall bias in survey responses. To limit social desirability, the survey was completely anonymous with no identifiers. In addition, the survey was worded such that the respondents could not be identified in any way and so that the questions would not be leading. To minimize recall bias, the survey was given to the undergraduates during the semester that they were enrolled in the research series (lines 90-94). There is a possibility of recall bias in alumni responses given that they may have completed the research series at any time within the timespan of 1988-2016, and we have discussed this limitation in the discussion section (lines 367-370). 

3. Related to the above, I wonder how students/alumni academic outcomes and demographics are related to response rates and survey answers. Are there subgroups of the UNK population that are heavily represented in the survey results?

The UNK student body is primarily white as described in our methods section (under the subsection “Department history and demographics”) and respondents predominantly identified as Caucasian in the surveys. Therefore, our survey results do heavily represent a white population. We have now also added a discussion regarding this in our discussion section. Specifically, we acknowledge that our survey results primarily represent a white population and we discuss the need to study undergraduate research experiences of students and alumni of diverse populations (lines 359-362). 

4. Authors mention the use of Fishers Exact Test to answer research questions. Elaborate on why you choose this method over Chi-square test, specially giving your sample size.

We have now added in our methods section an explanation regarding our choice to use the Fisher’s Exact Test for our analyses. The Fishers Exact Test was used because it is applicable for all sample sizes. Since we compared groups of a relatively small sample sizes (less than 1000), we needed to use a test that uses an exact procedure rather than a test such as a Chi-squared test that assumes a large sample size and therefore applies approximation (lines 103-106).

5. How was the survey constructed? Was there any psychometric analysis regarding validity or reliability of the measure? How is this related to different interpretations to some items by the respondents?

We have included in our methods section how the survey was constructed. The survey was constructed based on questions that the department wanted to gather student responses on. The survey was also independently validated by the IRB and faculty from the Psychology department at UNK (lines 98-101). 

6. The authors mentioned how future studies should be directed towards determining the attributes of mentors that make them good role models. On this note, was there any attempt to account for mentor heterogeneity? (other than gender). If not, how can attributes of mentors influence these results? Is this a potential limitation to this study?

In this study we only focussed on mentor gender and did not account for any other heterogeneity within the mentor population. It is possible that other attributes of mentors, aside from their gender may have played a role in how students responded to questions about their mentors. We acknowledge that this may be a potential limitation in our study and that future studies will require accounting for mentor heterogeneity by collecting more information about mentors other than just their gender (lines 364-367)

Conclusion & Discussion:

7. It appears that the undergraduate research experience at UNK was viewed as useful -according to authors analyses. How can these results be of any help to other institutions? What are the main elements of this research experience that added value? Are they any different to other mentoring programs in STEM? Can or should this be replicated in other institutions? If so, what are the authors recommendations.

We have now included in our discussion a section explaining the utility of our results to other institutes. We have elaborated on what we feel are the most valuable elements of the research program at out institute and also our recommendations for what can be replicated at other institutes (370-378).

8. It would be important to address the limitations of the study. Are these results generalizable to other contexts or institutions? Or even to other departments within UNK?

In our discussion we have added a section relating to the limitations of our study and addressed if our results are generalizable to other contexts or institutions. We have acknowledged that our study is based on the results from one department within one institution and have explained how our results can still be applicable and useful to other undergraduate institutes (lines 359-378).

9. I would suggest further developing ideas in the conclusion section. How are these results pertinent to this specific journal audience? What did the authors learn about phenomenon through analyzing data that can contribute to the existing literature?

We have now expanded the conclusions section to address how our results are pertinent to the audience of PLOS ONE and we have explained what we learned through analyzing the data, and how it contributes to the existing literature (lines 381-403).

---

## [Decision Letter · Decision Letter 1]

15 Nov 2021

Female mentors positively contribute to undergraduate STEM research experiences

PONE-D-21-21231R1

Dear Dr. Moghe,

We’re pleased to inform you that your manuscript has been judged scientifically suitable for publication and will be formally accepted for publication once it meets all outstanding technical requirements.

Kind regards,

Juan Cristobal Castro-Alonso, Ph.D.

Academic Editor

PLOS ONE

Additional Editor Comments (optional):

Reviewers' comments:

Reviewer's Responses to Questions

**Comments to the Author**

1. If the authors have adequately addressed your comments raised in a previous round of review and you feel that this manuscript is now acceptable for publication, you may indicate that here to bypass the “Comments to the Author” section, enter your conflict of interest statement in the “Confidential to Editor” section, and submit your "Accept" recommendation.

Reviewer #1: All comments have been addressed

2. Is the manuscript technically sound, and do the data support the conclusions?

Reviewer #1: Yes

3. Has the statistical analysis been performed appropriately and rigorously? 

Reviewer #1: Yes

4. Have the authors made all data underlying the findings in their manuscript fully available?

Reviewer #1: Yes

5. Is the manuscript presented in an intelligible fashion and written in standard English?

Reviewer #1: Yes

6. Review Comments to the Author

Reviewer #1: Thank you again for the opportunity to review this interesting article. The authors have carefully considered and satisfactorily addressed all the comments raised in the previous revision round. Particular attention was paid to acknowledging and discussing the limitations of the study. I recommend the publication of this manuscript. Congratulations!

7. PLOS authors have the option to publish the peer review history of their article (what does this mean?). If published, this will include your full peer review and any attached files.

Reviewer #1: No

---

## [Editor Report · Acceptance letter]

22 Nov 2021

PONE-D-21-21231R1 

Female mentors positively contribute to undergraduate STEM research experiences 

Dear Dr. Moghe:

I'm pleased to inform you that your manuscript has been deemed suitable for publication in PLOS ONE. Congratulations! Your manuscript is now with our production department. 

Kind regards, 

on behalf of

Dr. Juan Cristobal Castro-Alonso 

Academic Editor

PLOS ONE